# Mediating role of learned helplessness' components in the association between health literacy/social support and self-management among maintenance haemodialysis patients in Changsha, China: a cross-sectional study

Chunyan Xie [1,2] Li Li [1,3] Lin Zhou,[1,4] Cuifang Sun,[1,4] Yini Zhang,[1,4] Yamin Li [1]

CX and LL contributed equally.

For numbered affiliations see end of article.

**Correspondence to**
Professor Yamin Li;
aminny@csu.edu.cn

## ABSTRACT

**Objectives** To explore the multiple mediating roles of the learned helplessness's core system in the relationship between health literacy/social support and self-management.

**Design** Cross-sectional survey design.

**Setting** Changsha, China.

**Participants** 239 Chinese maintenance haemodialysis (MHD) patients.

**Methods** Two multiple mediator models were constructed based on the COM-B (Capacity, Opportunity, Motivation - Behaviour) model. A total of 239 Chinese MHD patients participated in a cross-sectional study, which included surveys on the Learned Helplessness Scale for MHD patients, Dialysis Knowledge Questionnaire, Social Support Scale and Self-Management Scale for Haemodialysis. The PROCESS macro in SPSS was used for mediated effects analysis.

**Results** Helplessness and internality partially mediated the relationship between health literacy/social support and self-management ((β=−0.212, p<0.01; β=0.240, p<0.01)/(β=−0.331, p<0.001; β=0.376, p<0.001)). The mediation effect size was 0.780 (95% CI (0.373 to 1.218)) in the health literacy model, accounting for 45.29% of the total effect, and 0.286 (95% CI (0.207 to 0.377)) in the social support model, accounting for 57.88% of the total effect. The differences in effect sizes for helplessness and internality in the two models were −0.080 (95% CI (−0.374 to 0.216)) and −0.041 (95% CI (−0.127 to 0.043)), respectively.

**Conclusion** Health literacy/social support directly affects MHD patients' self-management and indirectly affects it by changing learned helplessness, such as increasing internality while reducing helplessness.

## BACKGROUND

By 2017, approximately 700 million patients worldwide had chronic kidney disease (CKD), with China and India accounting for a significant portion (132 and 115 million,

## STRENGTHS AND LIMITATIONS OF THIS STUDY

⇒ The PROCESS macro in SPSS was used to verify the potential mediating effects of learned helplessness's components on the relationship between health literacy/social support and self-management in Chinese maintenance haemodialysis patients.

⇒ A culturally adapted and validated Chinese version of the Learned Helplessness Scale for maintenance haemodialysis patients was used.

⇒ Based on the ethnic diversity of China, participants could be recruited from a more diverse range of cultures and backgrounds.

⇒ Causality cannot be inferred from the current study, and future longitudinal studies are needed.

⇒ The self-reported data may pose a potential risk to the validity of the measurements.

respectively).[1] End-stage kidney disease (ESKD) refers to a complete or irreversible decline in renal function. Maintenance haemodialysis (MHD) is the most widely used alternative treatment in clinical practice.[2 3] MHD is defined as regular haemodialysis patients receiving for more than 3 months with a frequency of two to three times per week.[4] In China, most patients with ESKD are treated with MHD.[5]

Although MHD can prolong their lives, it cannot completely replace renal function. Patients must also slow their disease progression through self-management, such as long-term medications, restricted diet, fluid intake, regular exercise and arteriovenous fistula care.[6 7] Curtin *et al*[8] showed that self-management behaviours positively impacted MHD patients' overall health status. However, a large number of studies point to the generally unsatisfactory self-management status of

MHD patients. For example, less than 40% of them can strictly follow the doctor's advice to control their diet and water.[9] The overall level of Chinese MHD patients' self-management is low, with a score of 45.21±13.44 (total scale score of 80).[10] Therefore, it has become a priority for healthcare staff to investigate the controllable factors influencing MHD patients' self-management to improve their health status and quality of life.

## Health literacy/social support and self-management

A growing body of research has focused on how to improve MHD patients' self-management behaviours. Health literacy is an important influencing factor for improving self-management behaviours,[11] which reflects the degree of patients' awareness of the disease and determines their motivation and ability to acquire, understand and use knowledge to improve their health.[12] Notably, nursing education aims to improve patients' perceptions of disease and overall health literacy to induce changes in their behaviours.[13] When patients are more aware of their disease and health, they become aware of the importance of self-management and are more likely to adopt healthy behaviours, as confirmed by more studies. For example, Devraj et al[14] showed that health literacy in patients with CKD was positively associated with self-management, and a randomised controlled trial in Japan confirmed similar results.[11] Wu et al[15] found that health literacy could influence patients with CKD's self-management through the mediating role of self-efficacy, which is consistent with Hu et al.[16]

Second, social support is a factor that significantly impacts the treatment of most patients with chronic diseases.[17] It is a multidimensional concept that refers to the spiritual and material support given to individuals by various parties, such as family, relatives, friends and social organisations.[18] Adequate social support can reduce MHD patients' financial or psychological burdens and enable them to manage life better.[19] Several studies[20,21] have shown that the lack of family support in MHD patients is followed by poor medication adherence. Song et al[22] used a structural equation modelling approach to confirm that social support directly affects MHD patients' self-management and, together with the sense of consistency, explained 69% of the variance in self-management behaviour. Been-Dahmen et al[23] noted through qualitative interviews that family, friends, professionals and colleagues also provide essential emotional support to kidney transplant patients. Thus, social support is crucial for MHD patients to achieve their health goals.

Although previous studies have confirmed the relationship between health literacy, social support and self-management, most studies have overlooked possible potential pathways for this relationship. Therefore, the study was dedicated to exploring the multiple potential possibilities between health literacy/social support and self-management.

## Learned helplessness

Seligman first observed learned helplessness (LH) in 1967.[24] LH is described as a psychological state of powerlessness or self-abandonment when the individual perceives a disconnection between behaviour and outcome, manifested by cognitive deficits, decreased motivation and emotional maladjustment.[25,26] LH is often used to explain the phenomenon of diminished or absent motivation in individuals and extends to health-related problems.[27] For example, helpless patients adhere significantly less to regular treatment and seem to have more difficulty carrying out physicians' recommendations for diet, water and exercise.[28,29] LH was divided into two core components. Helplessness is the individual's perception that their behaviour cannot impact the outcome.[30] Internality refers to the individual's perception that they believe that they are the most dominant force dominating their life.[31]

It has been suggested that LH is associated with low social support and health literacy. First, adequate social support gives patients more tangible resources for life and intangible resources for spirituality, which can buffer the multiple negative emotions associated with illness.[32] Those with high social support have access to disease-related knowledge through multiple channels to reduce helplessness when coping with the disease. Notably, family support always plays a vital role in reducing patients' LH.[33] Smallheer et al[34] suggested that emotional support from family and friends, access to disease-related knowledge and trust and support with fellow patients and healthcare staff can help prevent patients' helplessness. In addition, it has been suggested that social support and health literacy can help maintain physical and mental harmony and influence helplessness by integrating them.[35]

## Theoretical basis

The Capacity, Opportunity, Motivation - Behaviour (COM-B) model, proposed by Michie et al, provides a framework for understanding individual behaviour change.[36] According to this model, although people with MHD are willing to engage in self-management, they give little action, which may be closely related to their ability, opportunity and motivation. Capacity refers to individuals' knowledge and skills to engage in behaviour change.[37] Opportunity refers to the external factors that promote behaviour change.[37] Motivation refers to the beliefs that guide an individual's behaviour change.[37] In this study, the good self-management exhibited by the patients first required a certain knowledge and skills of kidney disease care, that is, dialysis-related health literacy. At the opportunity level, social support from family members, friends and healthcare providers gives emotional and material security to patients. However, chronic LH can undermine patients' motivation to engage in self-management and affect their proactive health behaviours.

After a literature review and theoretical analysis, this study finally developed a mediating model with two core systems of LH as motivational factors, using social support

as an opportunity factor, health literacy as a competence factor and self-management as a dependent variable (as shown in online supplemental figure 1).

## The current study

There may be a relationship between health literacy/social support, LH and self-management behaviours. However, the underlying mechanisms between these relationships remain to be further identified. In this study, we developed two multiple mediator models based on the COM-B model to help us understand the role of LH in forming and changing health behaviours. To test this, we made the following hypotheses.

Hypothesis 1 (H1): Health literacy/social support can influence two dimensions of LH (helplessness and internality).

Hypothesis 2 (H2): Two dimensions of LH (helplessness and internality) predict self-management.

Hypothesis 3 (H3): Two dimensions of LH (helplessness and internality) parallelly mediate the relationship between health literacy/social support and self-management.

## METHODS
### Study design and participants

This is a cross-sectional study to investigate the association between health literacy/social support and self-management through the mediating role of the component of lean helplessness. This study followed the Strengthening the Reporting of Observational Studies in Epidemiology Checklist for cross-sectional studies (STROBE 2007, online supplemental file 1) guidelines to ensure the quality and specification of study reporting.

We used convenience sampling to select patients from a comprehensive Grade 3A hospital in Changsha, Hunan province, China, who met the following criteria: (1) being treated with haemodialysis for ≥3 months, (2) being informed and agreeing to participate in this study and (3) being conscious, being literate and having communication abilities. We excluded patients with mental illness, cognitive dysfunction or serious complications. The purpose and significance of this study were clarified to the participants before the survey to obtain informed consent. The questionnaires were administered 1 hour after the start or 1 hour before the end of dialysis because the patients' blood circulation was more stable. Each participant took approximately 10–15 min to complete the questionnaire.

### Measures
#### Health literacy

We used the Dialysis Knowledge Questionnaire developed by Curtin et al[8] to measure MHD patients' health literacy. The questionnaire was divided into five dimensions after cross-cultural adaptation by Li.[38] It includes anaemia, diet and medication, kidney function, dialysis and rehabilitation, with 24 items. The patients were asked to judge whether the items' descriptions were true, scoring 1 for a correct answer and 0 for an incorrect/unknown answer. The total score was 24. The higher the score, the higher the patient's knowledge of dialysis. Its Content Validity Index was 0.94, and the Cronbach's alpha coefficient was 0.701.

#### Social support

The Social Support Rating Scale developed by Xiao[39] was selected for this study. The scale consists of 10 items involving three dimensions: objective support, subjective support and support usage. Items 1–4 and 8–10 were scored on a 4-point Likert scale. Item 5 has five subitems, each of which has four options from 'none' to 'fully support', scoring 1–4 points. The scores of items 6 and 7 are calculated according to the number of support sources. The total score ranges from 12 to 66, with higher scores indicating higher levels of social support. The Cronbach's alpha coefficient for the total scale was 0.896. It has been used in Chinese renal dialysis patients.[40]

#### Self-management

The Self-Management Scale for Haemodialysis, developed by Song[41] in 2009, was used to measure MHD patients' self-management behaviours in this study. It has 20 items and is divided into four dimensions: problem solving, performing self-care, partnership and emotional processing. The scale was scored on a 4-point Likert scale, with scores 1–4 representing the frequency of the behaviour, that is, never, occasionally, often and always, respectively. The total score was 20–80. The higher the score, the better the patient's self-management behaviour. It was revised by Chinese Li et al, and its Cronbach's alpha coefficient was 0.813.[42]

#### Learned helplessness

There is no special scale for MHD patients' LH. Considering this limitation, we formed a tool called LHS-MHD-C based on Nicassio's Arthritis Helplessness Index scale.[43] Under his authorisation, we conducted cross-cultural adaptation, psychological measurement and evaluation. The LHS-MHD-C has 11 items and was classified into two dimensions (helplessness/internality) based on the results of exploratory factor analysis and confirmatory factor analysis, in line with the LH theory. It used a 5-point Likert scale with response options ranging from 1=strongly disagree to 5=strongly agree. Six items in internality are reverse scored. The higher the total score, the more severe the LH. In our previous study, the scale showed good internal consistency (Cronbach's α of 0.759), retest reliability (n=30, intragroup correlation=0.772) and split-half reliability (0.774).[44]

### Statistical analysis

According to numerous similar studies, the sample size of this study was initially determined by the observed variables of the theoretical model (ie, the ratio of sample size to observed variables should be at least between 10:1 and 15:1).[45] Additionally, the formula[46] for proportional

variables is $n=(Z^2P(1-P)]/d^2$, where Z is the statistic at a 95% confidence level (1.96), P is the proportion of the target population (0.1081) and d is the tolerable error of 0.05. Professional literature indicates that the minimum sample size for testing an mediator model is 200.[45] Taking into account a 20% sample loss rate, a total of 239 samples in this study would meet the minimum requirement.

Questionnaires with more than 10% missing values were excluded. IBM SPSS V.25 was used to complete the descriptive statistics and correlation analysis of all variables. If the data conformed to a normal distribution, the Pearson correlation coefficient was used to express the correlation between the two variables. The PROCESS macro in SPSS, developed by Hayes, was used for multiple mediation analyses. We adopted a bootstrap method (sampling was repeated 5000 times) to construct a 95% CI for significance testing of mediating effects.

This study employed three regression models to investigate the role of helplessness and internality as parallel mediators in the relationship between health literacy/social support and self-management. Model 1 examined the influence of health literacy/social support on helplessness. Model 2 explored the impact of health literacy/social support on internality. Model 3 examined the combined effects of health literacy/social support, helplessness and internality on self-management. Control variables such as age and gender were included as covariates in the models.

### Patient and public involvement

Patients and/or the public were not involved in the design, conduct, reporting or dissemination plans of this research.

### RESULTS
### Demographic characteristics

The study population consisted of 239 Chinese MHD patients aged 21–86 with a mean age of 55.61 and a SD=13.31. Among them, 59.8% were men. The characteristics of the participants are shown in table 1.

### Preliminary analysis

The correlations between the variables are listed in table 2. First, social support, health literacy and self-management were all negatively correlated with helplessness (r=−0.361, r=−0.599 and r=−0.463). Second, internality was positively associated with social support, health literacy and self-management, with correlation coefficients of 0.445, 0.691 and 0.489, respectively. Third, social support and health literacy positively predicted self-management (r=0.48, r=0.636).

### Parallel multiple mediator models

This study used three regression models to examine the parallel mediators' role of helplessness and internality (table 3 and figure 1). The results of Model 1 showed that health literacy/social support was negatively associated

**Table 1** Descriptive statistics of sociodemographic characteristics (n=239)

| Variable | | n | % |
|---|---|---|---|
| Gender | | | |
| | Male | 143 | 59.8 |
| | Female | 96 | 40.2 |
| Age (years) | | | |
| | 18–44 | 46 | 19.2 |
| | 45–59 | 100 | 41.8 |
| | ≥60 | 93 | 38.9 |
| Education level | | | |
| | Elementary school | 26 | 10.9 |
| | Secondary school | 126 | 52.7 |
| | University and above | 87 | 36.4 |
| Employed status | | | |
| | On duty | 10 | 4.2 |
| | Part-time job | 37 | 15.5 |
| | Off duty | 192 | 80.3 |
| Protopathy | | | |
| | Hypertension | 52 | 21.8 |
| | Diabetes | 48 | 20.1 |
| | Chronic nephritis | 105 | 43.9 |
| | Polycystic kidney | 20 | 8.4 |
| | Others | 14 | 5.9 |
| Duration of the maintenance haemodialysis (years ) | | | |
| | <1 | 54 | 22.6 |
| | 1–3 | 42 | 17.6 |
| | 3–7 | 72 | 30.1 |
| | ≥7 | 71 | 29.7 |

with helplessness (β=−0.616, p<0.001; β=−0.354, p<0.001). In model 2, health literacy/social support was positively associated with internality (β=0.666, p<0.001; β=0.416, p<0.001). In Model 3, we found that in the health literacy model, the effects of both helplessness and internality on self-management were significant (β=−0.212, p<0.01; β=0.240, p<0.01), and health literacy remained a significant positive predictor of self-management (β=0.351, p<0.001). Similarly, in the social support model, social support, helplessness and internality all had significant effects on self-management (β=0.199, p<0.001; β=−0.331, p<0.001; β=0.376, p<0.001). Therefore, we deduced that helplessness and internality partially mediate the relationship between health literacy/social support and self-management.

To confirm the significance of the model's total, direct and indirect effects, we used a bootstrap analysis with 5000 repeated sampling tests, and the CI was set at 95%. The results showed significant parallel mediating effects of helplessness and internality on the relationship between

**Table 2** Descriptive statistics and Pearson's correlation coefficients for all variables of interest (n=239)

| Variable | M | SD | 1 | 2 | 3 | 4 | 5 |
|---|---|---|---|---|---|---|---|
| 1. Helplessness | 16.858 | 4.70 | 1 | | | | |
| 2. Internality | 17.351 | 6.411 | -0.140* | 1 | | | |
| 3. Social support | 44.628 | 10.094 | -0.361** | 0.445** | 1 | | |
| 4. Self-management | 53.197 | 10.537 | -0.463** | 0.489** | 0.480** | 1 | |
| 5. Health literacy | 16.889 | 3.921 | -0.599** | 0.691** | 0.443** | 0.636** | 1 |

Note: *p<0.05, **p<0.01.
M, mean.

health literacy/social support and self-management. The indirect effect size of health literacy via helplessness and internality on self-management was 0.780 (95% CI (0.373 to 1.218)), accounting for 45.29% of the total effect. The indirect effect size of social support via helplessness and internality on self-management was 0.286 (95% CI (0.207 to 0.377)), accounting for 57.88% of the total effect. In addition, the differences in effect sizes for helplessness and internality in the two models were –0.080 (95% CI (–0.374 to 0.216)) and –0.041 (95% CI (–0.127 to 0.043)), respectively. This result indicated that helplessness and internality did not differ significantly in the mediated effect size of the health literacy/social support and self-management relationship model (table 4).

## DISCUSSION
The findings confirm the COM-B model's applicability in health-related behavioural research among MHD patients. First, health literacy/social support, as an important competence and opportunity factor for MHD patients, was positively and negatively associated with internality and helplessness, respectively, and positively associated with self-management. Second, LH, an important motivational factor influencing individual behaviour, and its two major systems, internality and helplessness, were

positively and negatively associated with self-management in MHD patients, respectively. Furthermore, our findings suggest that health literacy/social support influences self-management through internality and helplessness as mediating factors.

It is worth noting that internality, the core system of LH, has similarities with self-efficacy in that both can directly or indirectly influence self-management. However, the two are presented in different theoretical contexts, determining the intervention's core. Self-efficacy, derived from Bandura's social learning theory,[47] refers to an individual's subjective evaluation of his or her abilities, driven by personal expectations and goals. In this context, the core of self-management interventions is to help patients set reasonable goals and gradually develop their self-confidence. Internality originates from Rotter's locus of control theory,[31] which refers to a stable way of attributing the outcome of events internally after individuals have learned through experience. The core of its health behaviour intervention is to guide patients to rational attribution, reduce helplessness and enhance motivation.

### Correlation between health literacy/social support and LH
This study first verified the correlation between health literacy and the component of LH (internality/helplessness). According to the knowledge-attutide-practice

**Table 3** Results of the multiple mediation analysis

| Regression model | | Fitting indices | | Regression coefficient | |
|---|---|---|---|---|---|
| Outcome variable | Predictor variable | R² | F | β | t |
| Helplessness | Social support | 0.402 | 6.357 | -0.354 | 5.808*** |
| Internality | Social support | 0.523 | 12.422 | 0.416 | 7.337*** |
| Self-management | Social support | 0.668 | 20.526 | 0.199 | 3.411*** |
| | Helplessness | | | -0.331 | 6.172*** |
| | Internality | | | 0.376 | 6.522*** |
| Helplessness | Health literacy | 0.629 | 21.630 | -0.616 | 11.682*** |
| Internality | Health literacy | 0.722 | 35.981 | 0.666 | 14.197*** |
| Self-management | Health literacy | 0.671 | 20.828 | 0.351 | 3.635*** |
| | Helplessness | | | -0.212 | 2.998** |
| | Internality | | | 0.240 | 3.028** |

Note: **p<0.01, ***p<0.001.

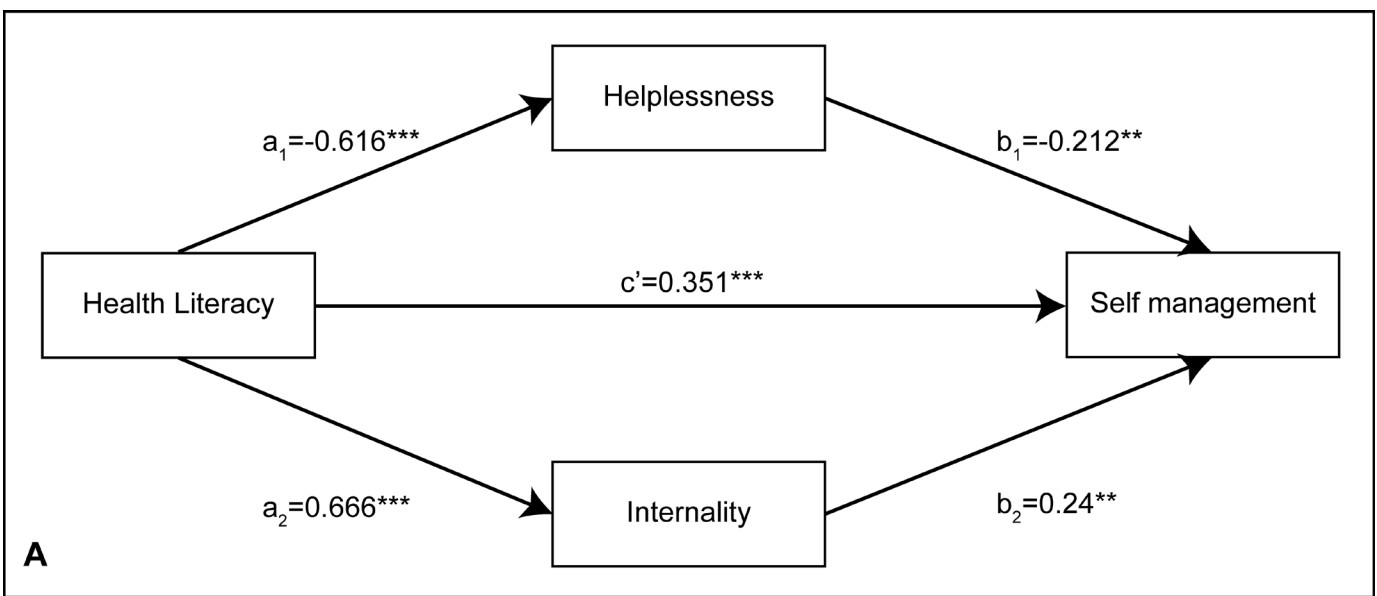

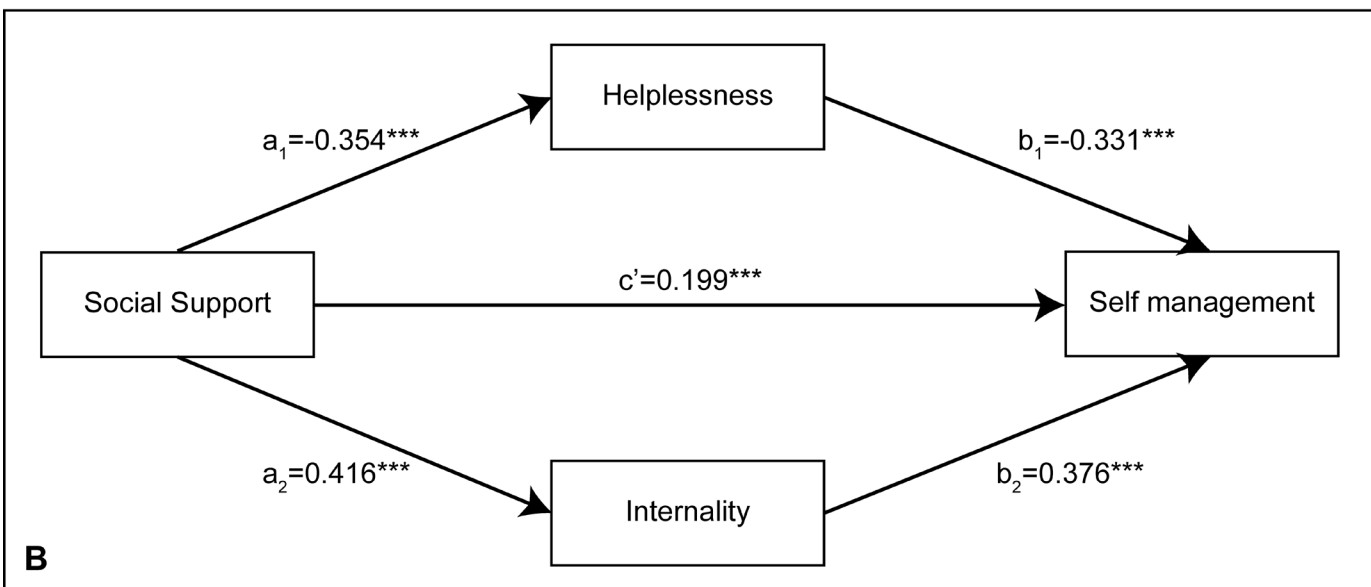

**Figure 1** Parallel multiple mediator models. (A) Health literacy model and (B) social support model.

theory,[48] patients with some knowledge of kidney disease and dialysis rated their self-behaviour better and had stronger beliefs about their illness. They could make rational judgements and perceptions when encountering negative disease-related events, accompanied by a lower sense of helplessness, which was verified in this study. Conversely, patients with higher internality are more likely to acquire disease-related knowledge through various means, which is vital for patients' self-management. Those patients who were chronically helpless lacked the motivation to consistently learn self-care knowledge and skills. This point echoes the findings of our previous qualitative study.[49] Therefore, successful renal disease education should provide knowledge and increase motivation. Healthcare professionals should also emphasise patient internality and empowerment to help patients maintain a positive mindset and long-term intrinsic motivation to face their disease.

Next, this study verified the close relationship between internality/helplessness and social support. According to Shahriari et al,[50] social support, as an important external resource, can lead to significant psychological changes in patients. This study reflects two critical variables of internality and helplessness. On the one hand, a lack of social support can induce helplessness in patients, mainly depression, social avoidance and abandonment. On the other hand, adequate social support can help patients expand access to social resources, increase internality in coping with illness and reduce helplessness, which is similar to the findings of Wang et al[51] and Wu.[52] It is worth noting that Jansen et al[53] noted that excessive support given to MHD patients may affect their internality. In this

**Table 4** Bootstrap analysis of multiple mediation effects

| | Effect size | SE | LLCI | ULCI |
|---|---|---|---|---|
| X1 (social support) | | | | |
| Total | 0.494 | 0.060 | 0.375 | 0.612 |
| Direct | 0.208 | 0.061 | 0.088 | 0.328 |
| Total indirect | 0.286 | 0.043 | 0.207 | 0.377 |
| Indirect effect (M1=helplessness) | 0.122 | 0.030 | 0.068 | 0.186 |
| Indirect effect (M2=internality) | 0.163 | 0.032 | 0.108 | 0.230 |
| (C1) | -0.041 | 0.043 | -0.127 | 0.043 |
| X2 (health literacy) | | | | |
| Total | 1.723 | 0.139 | 1.449 | 1.996 |
| Direct | 0.942 | 0.259 | 0.432 | 1.453 |
| Total indirect | 0.780 | 0.217 | 0.373 | 1.218 |
| Indirect effect (M1=helplessness) | 0.350 | 0.119 | 0.127 | 0.599 |
| Indirect effect (M2=internality) | 0.430 | 0.143 | 0.161 | 0.720 |
| (C1) | -0.080 | 0.148 | -0.374 | 0.216 |

Note: C1=helplessness minus internality.
LLCI, Lower Level Confidence Interval; ULCI, Upper Level Confidence Interval.

regard, assisting patients to build an appropriate social support environment is essential to enhance their internality and temporarily escape from helplessness.

### Relevance of LH to self-management

According to Seligman *et al*, LH is essential to human behaviour.[54] The present study first verified that helplessness is a negative predictor of self-management behaviour, a result similar to other studies.[55–57] Helplessness plays an important cognitive, motivational and behavioural role in individuals. It can influence patients' attitudes toward illness and treatment and determine their health-related behaviours and persistence.[58] MHD patients feel isolated due to worsening kidney function, complications and death anxiety. In addition, anxiety, panic, depression, regret and social isolation are all feelings that contribute to the emotional burden of MHD patients. Many accept their condition with a sense of helplessness—a feeling that whatever they do will not have an impact on the future. Overwhelming helplessness may weaken patients' motivation to engage in treatment. This sense of helplessness may also be reinforced by recurrent experiences of uncontrollable symptoms, which may be deeply rooted in the background of living with the situation.

Next, we verified that internality is a positive predictor of self-management, which is consistent with other studies.[59 60] Numerous scholars have argued that control is one of the basic psychological needs of humans, determined by humans' social attributes. Internality

determines the degree to which an individual can grasp stressful events. When patients have high internality, they can actively use various resources and establish a more positive way of coping with their illness. In this study, MHD patients with higher internality tended to perceive the existing treatment outcome as a result of their behaviour. For example, for adverse events in disease treatment, patients more often attributed them to their lack of effort and believed they could mitigate the effects of adverse events by changing their behaviours.

### The mediating role of LH

The results showed that health literacy/social support could influence the self-management of MHD patients through the mediating role of helplessness and internality, similar to other studies.[58 61] First, health literacy reflects patients' perceptions of illness and treatment and can determine their ability and motivation to take positive steps to improve their health. Patients with higher health literacy have a higher ability to access medical information and communicate with medical staff and are better able to make rational judgements independently when faced with changes in their health status, thereby reducing feelings of helplessness. Individuals with a correct understanding of the disease can influence their emotional responses. When MHD patients have more knowledge about kidney disease and healthcare, their ability to perceive disease deterioration is enhanced, and their sense of control over the disease increases. Based on this, patients can more actively and purposefully engage in their health behaviour.

Social support plays a crucial role as an external resource, helping to buffer the effects of stressors. MHD patients often face challenges in disease control, the absence of social support, can contribute to feelings of helplessness.[62] Helplessness can lead patients to believe that they have limited control over their situation, attributing outcomes to external factors like fate, social background or external forces. This external locus of control reduces their initiative and leads to self-abandonment, social avoidance and decreased trust in others. On the other hand, internality, combined with social support, creates a favourable internal and external environment for patients, providing them with resources and enhancing their confidence in controlling their disease outcome. This increased confidence in turn motivates patients to actively engage in health-related behaviours.

### Implications for future research

The results confirmed that helplessness and internality are important ways for health literacy/social support to influence self-management. The results help us understand the mechanisms of possible changes in the health-related behaviours of MHD patients and provide theoretical support for future studies. In the future, healthcare providers can focus on LH as an intervention target and identify and assess the level of LH as early as possible while providing health education and medical support

for MHD patients. Patients can be empowered by encouraging or rewarding their self-management behaviours to improve their internality and reduce helplessness.

Abramson *et al*[63] showed that individual helplessness also triggers a group's helplessness, harming individual behaviour. Most MHD patients in China have wardmate groups, and LH events among peers may impact other patients' behaviours. Future studies may investigate the interaction of LH in groups on MHD patients' behaviours.

## Limitations and future directions

First, we selected dialysis patients from a southern city in China by convenience sampling, which may limit the generalisability of the results. Based on the ethnic diversity of China, future studies could recruit participants from more diverse cultures and backgrounds. Second, causality cannot be inferred from the current study, and future longitudinal studies are needed. Third, the reliance on self-reported data may pose a potential risk to the validity of the measurements.

## CONCLUSION

This study sheds light on how health literacy and social support impact self-management in MHD patients. It emphasises the role of LH in driving behavioural changes. The findings provide valuable insights for developing intervention strategies to enhance health-related behaviours among MHD patients. Clinical caregivers should prioritise preventing, identifying, and monitoring LH during health education and medical support.

**Author affiliations**
[1]Clinical Nursing Teaching and Research Section, The Second Xiangya Hospital of Central South University, Changsha, China
[2]Xiangya Nursing School, Central South University, Changsha, China
[3]Department of Urology, Xinjiang Medical University Affiliated First Hospital, Urumqi, China
[4]Blood Purification Center, Department of Nephrology, The Second Xiangya Hospital of Central South University, Changsha, China

**Acknowledgements** We express our heartfelt gratitude to all members who assisted in the data collection and processing and to every maintenance haemodialysis patient who participated in this study.

**Contributors** CX and LL contributed to the conceptualisation, formal analysis and writing the original draft. CS and YZ made contributions to the data curation and verification. YL and LZ contributed to the writing, review and editing. YL acts as a guarantor for the final manuscript.

**Funding** This research received grants from the National Natural Science Foundation of China (No.81873806), Major Scientific and Technological Projects in Hunan Province (No.2020SK2085), 2022 Hunan Province Postgraduate Scientific Research Innovation Project (No.12300-150110021) and 2022 Central South University Postgraduate Independent Exploration and Innovation Program (No. 1053320213471).

**Competing interests** None declared.

**Patient and public involvement** Patients and/or the public were not involved in the design, or conduct, or reporting, or dissemination plans of this research.

**Patient consent for publication** Not applicable.

**Ethics approval** This study involves human participants and was approved by Ethics Committee of Xiangya Nursing School of Central South University (No.

E202255). Participants gave informed consent to participate in the study before taking part.

**Provenance and peer review** Not commissioned; externally peer reviewed.

**Data availability statement** Data are available upon reasonable request.

**ORCID iDs**
Chunyan Xie http://orcid.org/0000-0002-6247-4442
Li Li http://orcid.org/0000-0002-6569-6008
Yamin Li http://orcid.org/0000-0002-8396-5208

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
