## [Reviewer comments · BMJ Open]

ARTICLE DETAILS

TITLE (PROVISIONAL)	The mediating role of learned helplessness's components in the association between health literacy/social support and self-management among maintenance hemodialysis patients in Changsha, China: a cross-sectional study
AUTHORS	Xie, Chunyan; Li, Li; Zhou, Lin; Sun, Cuifang; Zhang, Yini; Li, Ya-Min

VERSION 1 – REVIEW

REVIEWER	Bayuo, Jonathan Hong Kong Polytechnic University, School of Nursing
REVIEW RETURNED	31-Jan-2023

GENERAL COMMENTS	The authors present a cross-sectional study regarding the mediating role of learned helplessness in the association between health literacy/ social support and self-management. Please see the comments below for your consideration: Title: The authors may need to consider including "maintenance haemodialysis patients" to give readers a clear start. As it stands, its only when one reads the abstract, that it is seen that the study focused on persons undergoing maintenance haemodialysis. Introduction: 1. Line 74: The overall level of Chinese MHD patients' self-management is low; is there a figure or range to indicate how low this was? Same applies to the subsequent statement.2. Lines 75-76: the authors note that effective implementation of self-management requires a combination of multiple factors but none of these are highlighted. At least, two or three factors specific to persons with kidney disease should be mentioned here.3. The introduction is extensive; a figure representing how these concepts may be related should be presented.4. Regarding the theoretical basis, several concepts are presented, but the authors decide to focus on three. There is however no justification why this is so. This should be considered to strengthen the statement. Methods 1. The STROBE guidelines are used for reporting but not mentioned in the methods section.2. The reliability scores of the learned helplessness are presented but not referenced (lines 219 - 220). Please add a reference. Results 1. A new variable "internality" is introduced here which appeared in the hypothesis as a component of learned helplessness, but not in
---

	the background. It will be helpful if this is explained in the background. The concept "learned helplessness" seems to be broad and need to be unpacked well to avoid confusion. Discussion 1. This section is well raised. However, how the findings link to the theoretical basis (COM-B model) remains poorly articulated. Please revise accordingly.
--	---

REVIEWER	Dunn, Patrick American Heart Association Inc, Center for Health Tech and Innovation
REVIEW RETURNED	18-Feb-2023

GENERAL COMMENTS	Thank you for your work in this very important topic. Here are my comments. You do not mention MHD or chronic kidney disease in your title. It might be helpful as people look for work in this area that your title be more specific. How did you determine the sample size of 239? Was a power analysis conducted? Is the sample large enough to do a mediation analysis? Is your sample representative of the CKD/MHD population? Since you are doing a correlation analysis based on the surveys it would be very helpful to see the descriptives of the surveys.
--

REVIEWER	Guo, Shuaijun University of Melbourne School of Population and Global Health
REVIEW RETURNED	20-Feb-2023

GENERAL COMMENTS	Thank you for the opportunity to review this interesting paper, which examines the mediating role of helplessness and internality in the relationship between two exposures (health literacy and social support) and self-management behaviours. The authors used multiple mediation analyses, which are appropriate to answer the research questions. However, there are several concerns that need to be addressed:  1. Covariates: The proposed model (Figure 1) is theoretically supported, which is great. However, there are also covariates (e.g., age, gender, education level) that may bias the results. The authors need to control for these covariates in the results of Table 3 and Table 4. 2. Correlation: What correlation did the authors use (Pearson or Spearman)? Have the authors checked the normality distribution before conducting the correlation analysis? If yes, please add. 3. Missing data: How are missing data addressed? Did the authors use the complete case dataset? For example, as for the 24-item health literacy scale, if the respondent only answered 20 items, how to calculate the total score? This should be specified in the "Statistical analysis" section. 4. Re-structure the statistical methods: Some information on the multiple mediation analyses in the Results section should be moved upfront in the "Statistical analysis" section. Sth like "We used
---

	three-step regression models to examine Model 1 is to ...; Model 2 is to ...; Model 3 is to ...” Some other comments are listed here:  • Page 8 line 122-123: “It has been suggested that learned helplessness is associated with high social support and health literacy” should be “It has been suggested that learned helplessness is associated with low social support and low health literacy.” • Page 11 Line 196-197: “This instrument has been used in Chinese renal dialysis patients.” Reference is needed at the end. • Page 12 Line 218 : “In our previous study,” reference is needed here. • Table 1 on Page 14: The ethnicity variable is missing here. The authors highlight “ethnicity” is an important variable in the Limitation section. Why? Does this mean only “Han” ethnicity was recruited? Is there any study that found ethnicity will influence the findings of relationship between health literacy/social support, helplessness and internality, and self-management behaviours? • Table 1 on Page 14: The education level does not include “high school”? Maybe change the “Middle school” to “Secondary school”? • Page 18 Line 294-316: The first paragraph in the Discussion section needs to be revised: It should be the key summary of findings, corresponding to the aforementioned three hypotheses. Also, there is lack of discussion on the hypothesis 1 in the Discussion section. Currently, the authors focuses on more discussions on hypothesis 2 and hypothesis 3. • Page 19 Line 309-313: The definitions of helplessness and internality (“Learned helplessness was divided into ... their life.”) should be moved upfront in the Introduction “Learned helplessness” section. • This study treated internality as one important mediating variable based on the learned helplessness theory, which is similar to self-efficacy (the author also mentioned a bit in the introduction) and locus of control. It would be good to add some discussion around their differences and roles in self-management. The authors already cited some good references (ref 6 “self-efficacy”; ref 16 “self-efficacy”; ref 51 “locus of control”; ref 53 “locus of control”).
--	--

VERSION 1 – AUTHOR RESPONSE

Reviewer: 1

Mr. jonathan bayuo, Hong Kong Polytechnic University

Comments to the Author:

The authors present a cross-sectional study regarding the mediating role of learned helplessness in the association between health literacy/ social support and self-management. Please see the comments below for your consideration:

Comment 1: Title: The authors may need to consider including "maintenance haemodialysis patients" to give readers a clear start. As it stands, its only when one reads the abstract, that it is seen that the study focused on persons undergoing maintenance haemodialysis.

Response 1: Thank you for your friendly advice. We have changed this title to make the research subject and research questions more understandable to our readers.

Change 1 (Title):

The mediating role of learned helplessness's components in the association between health literacy/social support and self-management among maintenance hemodialysis patients in Changsha, China: a cross-sectional study

Introduction:

Comment 2: 1. Line 74: The overall level of Chinese MHD patients' self-management is low; is there a figure or range to indicate how low this was? Same applies to the subsequent statement.

Response 2: Thank you for your comments. We have added the relevant data and its reference.

Change 2 (Page 4, line 74-76):

The overall level of Chinese MHD patients' self-management is low, with a score of 45.21 ± 13.44 (total scale score of 80)¹⁰

[10]10. Li H, Cao YD, Jiang YF, et al. Reliability and validity tests of Hemodialysis Self-Management Instrument. Chinese Journal of Nursing 2015;50(11):1392-95.

Comment 3: 2. Lines 75-76: the authors note that effective implementation of self-management requires a combination of multiple factors but none of these are highlighted. At least, two or three factors specific to persons with kidney disease should be mentioned here.

Response 3: Thank you for your comments. We have added several factors related to the self-management of dialysis patients and fine-tuned the original expressions for sentence coherence.

Change 3 (Page 4, line 76-78):

Effective implementation of self-management requires a combination of multiple factors, such as the patient's gender, age, dialysis duration, and frequency. However, these relevant factors are not easily changed.

Comment 4: 3. The introduction is extensive; a figure representing how these concepts may be related should be presented.

Response 4: Thank you for your suggestion. Figure 1 in the original manuscript maybe present the potential connection between these concepts.

Change 4 (Figure 1):

Figure 1. Theoretical framework based on COM-B model. HL: Health literacy; LH: Learned helplessness; SS: Social support; SM: Self-management.

Comment 5: 4. Regarding the theoretical basis, several concepts are presented, but the authors decide to focus on three. There is however no justification why this is so. This should be considered to strengthen the statement.

Response 5: Thank you for your suggestion. We have rewritten the theoretical basis section.

Change 5 (Page 7-8, line 144-163):

The Competence, Opportunity, and Motivation-Behavior (COM-B) model is a theoretical model related to individual behavior change proposed by Michie³⁵. The COM-B model explains that competence, opportunity, and motivation can influence an individual's behavior, whereas competence and opportunity can influence the individual's behavior indirectly through motivation³⁶. According to this model, although people with MHD are willing to engage in self-management, they give little action, which may be closely related to their ability, opportunity, and motivation. Capacity refers to individuals' knowledge and skills to engage in behavior change³⁶. Opportunity refers to the external factors that promote behavior change³⁶. Motivation refers to the beliefs that guide an individual's behavior change³⁶. In this study, the good self-management behaviors exhibited by the patients first required a certain knowledge and skills of kidney disease care, i.e., dialysis-related health literacy. At the opportunity level, social support from family members, friends, and healthcare providers gives emotional and material security to patients. However, chronic learned helplessness can undermine patients' motivation to engage in self-management and affect their proactive health behaviors.

After a literature review and theoretical analysis, this study finally developed a mediating model with two core systems of learned helplessness as motivational factors, using social support as an opportunity factor, health literacy as a competence factor, and self-management as a dependent variable. (As shown in Figure 1)

Methods

1. The STROBE guidelines are used for reporting but not mentioned in the methods section.

Response: Thank you for your suggestion. To make this study more transparent, we have added a description of the use of STROBE guidelines in the methods section and supplementary material.

Change (Page 9, line 184-187):

This study followed the Strengthening the Reporting of Observational Studies in Epidemiology Checklist for cross-sectional studies (STROBE 2007, Supplementary File 1) guidelines to ensure the quality and specification of study reporting.

2. The reliability scores of the learned helplessness are presented but not referenced (lines 219 - 220). Please add a reference.

Response: Thank you for your comment. We have added the relevant citation.

Change (Page 12, line 236-238):

In our previous study, the scale showed a good internal consistency [(Cronbach's α of 0.759), retest reliability (n=30, intragroup correlation=0.772), and split-half reliability [0.774]⁴⁴.

[44] Xie C, Zhou L, Sun C, et al. The cross-cultural adaptation and psychometric evaluation of a learned helplessness scale for maintenance hemodialysis patients in China (LHS-MHD-C). 2023

Results

1. A new variable "internality" is introduced here which appeared in the hypothesis as a component of learned helplessness, but not in the background. It will be helpful if this is explained in the background. The concept "learned helplessness" seems to be broad and need to be unpacked well to avoid confusion.

Response: Thank you for your suggestion. We have reinterpreted the meaning of learned helplessness and added a description of the internality in the background section.

Change

(Page 6, line 118-121): Seligman first observed learned helplessness (LH) in 1967²⁴. LH is described as a psychological state of powerlessness or self-abandonment when the individual perceives a disconnection between behavior and outcome, manifested by cognitive deficits, decreased motivation, and emotional maladjustment^{25 26}

(Page 6-7, line 127-129): Internality refers to the individual's perception that they believe that they are the most dominant force dominating their life³¹.

Discussion

1. This section is well raised. However, how the findings link to the theoretical basis (COM-B model) remains poorly articulated. Please revise accordingly.

Response: Thank you for your comments. In the discussion section, we have added a description of the relationship between the theoretical model and the results.

Change (Page 18, line 328-340):

Based on the COM-B model, this study jointly explored potential factors influencing self-management in MHD patients at three levels: opportunity, competence, and motivation, and elucidated the pathways among the factors by constructing a theoretical model. Our findings confirm the COM-B model's applicability in health-related behavioral research among MHD patients. First, health literacy/social support, as important competence and opportunity factor for MHD patients, was positively and negatively associated with internality and helplessness, respectively, and positively associated with self-management. Second, learned helplessness, an important motivational factor influencing individual behavior, and its two major systems, internality and helplessness, were positively and negatively associated with self-management in MHD patients, respectively. Furthermore, our findings suggest that health literacy/social support influences self-management through internality and helplessness as mediating factors.

Reviewer: 2

Dr. Patrick Dunn, American Heart Association Inc, Walden University

Comments to the Author:

Dear Authors,

Thank you for your work in this very important topic. Here are my comments.

Comment 1: You do not mention MHD or chronic kidney disease in your title. It might be helpful as people look for work in this area that your title be more specific.

Response 1: Thank you for your friendly advice. We have changed this title based on your suggestion.

Change 1(Title):

The mediating role of learned helplessness's components in the association between health literacy/social support and self-management among maintenance hemodialysis patients in Changsha, China: a cross-sectional study

Comment 2: How did you determine the sample size of 239? Was a power analysis conducted? Is the sample large enough to do a mediation analysis?

Response 2: Thank you for your suggestion. Initially, given the methodology used in many related studies (i.e., the ratio of a sample size to observed variables is at least between 10:1 and 15:1), the sample size for the study was determined by the observed variables of our theoretical model. Also, the 20% sample attrition rate has to be considered. On this basis, the sample size of this study is up to the standard.

However, we attach importance to the issue of the sample size test you raised. In response, we recalculated the sample size based on the relevant formula. For proportional variables, according to the formula $n = [Z^2 P(1-P)]/d^2$, where Z is the statistic at the confidence level, p is the proportion of the

target population, and d is the tolerance error of 0.05. In this study, the 95% confidence level Z has a statistic of 1.96, p is 0.1081, d is 5%, and the confidence level $1-\alpha=0.95$. The final sample size is calculated as 148. The professional books point out that the sample size of the mediated model test is at least 200, so the 293 samples in this study can meet the minimum requirements.

Change 2: N/A.

Comment 3: Is your sample representative of the CKD/MHD population?

Response 3: Thank you for your comment. We must acknowledge that the study used a convenience sampling method to select hemodialysis patients from a southern city in China and did not include populations other than Han Chinese. Therefore, the sample is not representative of the whole of China, which is one of the limitations of this study. Due to this circumstance, we have revised the title and added it to the Limitation in the Discussion section.

However, the results of this study have enriched the knowledge of the learned helplessness phenomenon in the dialysis setting to some extent.

Change 3:

(Title): The mediating role of learned helplessness's components in the association between health literacy/social support and self-management among maintenance hemodialysis patients in Changsha, China: a cross-sectional study

(Page 24, line 457-460) First, We selected dialysis patients from a southern city in China by convenience sampling, which may limit the generalizability of the results. Based on the ethnic diversity of China, future studies could recruit participants from more diverse cultures and backgrounds.

Since you are doing a correlation analysis based on the surveys it would be very helpful to see the descriptives of the surveys.

Response: Thank you for your comments. We have added the descriptive statistics results for the main variables in Table 2.

Change:

292 **Table 2** Descriptive statistics and Pearson's correlation coefficients for all variables
293 of interest (n=239).

Variable	M	SD	1	2	3	4	5
1.Helplessness	16.858	4.70	1				
2.Internality	17.351	6.411	-.140*	1			
3.Social Support	44.628	10.094	-.361**	.445**	1		
4.Self-Management	53.197	10.537	-.463**	.489**	.480**	1	
5.Health Literacy	16.889	3.921	-.599**	.691**	.443**	.636**	1

Note: * $p < 0.05$, ** $p < 0.01$. M=mean, SD=standard deviation.

294

Reviewer: 3

Mr. Shuaijun Guo, University of Melbourne School of Population and Global Health

Comments to the Author:

See attached

*Please see the attached report from this reviewer

Thank you for the opportunity to review this interesting paper, which examines the mediating role of helplessness and internality in the relationship between two exposures (health literacy and social support) and self-management behaviours. The authors used multiple mediation analyses, which are appropriate to answer the research questions. However, there are several concerns that need to be addressed:

Comment 1. Covariates: The proposed model (Figure 1) is theoretically supported, which is great. However, there are also covariates (e.g., age, gender, education level) that may bias the results. The authors need to control for these covariates in the results of Table 3 and Table 4.

Response 1: Thank you for your good suggestions. We have added the covariates of the Table 1 into the models and re-analyzed the data to obtain the new Tables 3 and 4. In addition, we have adjusted the textual expressions related to the data, such as the Statistical analysis, Results, and Abstracts section.

Change 1:

(Page 13, line 258-259)

Statistical analysis

.....Control variables such as age and gender were introduced in the model as covariates.

(Page 2, line 28-35)

Results: Helplessness and internality partially mediated the relationship between health literacy/social support and self-management [$(\beta = -0.212, P < 0.01; \beta = 0.240, P < 0.01) / (\beta = -0.331, P < 0.001; \beta = 0.376, P < 0.001)$]. The mediation effect size was 0.7802 (95% CI [0.373, 1.218]) in the health literacy model, accounting for 45.29% of the total effect, and 0.2857 (95% CI [0.207, 0.377]) in the social support model, accounting for 57.88% of the total effect. The differences in effect sizes for helplessness and internality in two models were -0.080 (95% CI [-0.374, 0.216]), -0.041 (95% CI [-0.127, 0.043]), respectively.

(Page 15-16, line 297-306): The result of Model 1 showed that health literacy/social support was negatively associated with helplessness ($\beta = -0.616, P < 0.001; \beta = -0.354, P < 0.001$). In model 2, health literacy/social support was positively associated with internality ($\beta = 0.666, P < 0.001; \beta =$

0.416, $P < 0.001$). In the Model 3, we found that in the health literacy model, the effects of both helplessness and internality on self-management were significant ($\beta = -0.212$, $P < 0.01$; $\beta = 0.240$, $P < 0.01$), and health literacy remained a significant positive predictor of self-management ($\beta = 0.351$, $P < 0.001$). Similarly, in the social support model, social support, helplessness, and internality all had significant effects on self-management ($\beta = 0.200$, $P < 0.001$; $\beta = -0.331$, $P < 0.001$; $\beta = 0.376$, $P < 0.001$).

(Page 17, line 315-321):The indirect effect size of health literacy via helplessness and internality on self-management was 0.780 (95% CI [0.373, 1.218]), accounting for 45.29% of the total effect. The indirect effect size of social support via helplessness and internality on self-management was 0.286 (95% CI [0.207, 0.377]), accounting for 57.88% of the total effect. In addition, the differences in effect sizes for helplessness and internality in the two models were -0.080 (95% CI [-0.374, 0.216]), -0.041 (95% CI [-0.127, 0.043]), respectively.

10

Table 3 Results of the multiple mediation analysis.

Regression Model		Fitting Indices		Regression Coefficient	
Outcome Variable	Predictor Variable	R ²	F	β	t
Helplessness	Social Support	0.402	6.357	-0.354	-5.808***
Internality	Social Support	0.523	12.422	0.416	7.337***
Self-management	Social Support	0.668	20.526	0.199	3.411***
	Helplessness			-0.331	-6.172***
	Internality			0.376	6.522***
Helplessness	Health Literacy	0.629	21.630	-0.616	-11.682***
Internality	Health Literacy	0.722	35.981	0.666	14.197***
Self-management	Health Literacy	0.671	20.828	0.351	3.635***
	Helplessness			-0.212	-2.998**
	Internality			0.240	3.028**

Note: * $p < 0.05$, ** $p < 0.01$, *** $p < 0.001$.

Table 4 Bootstrap analysis of multiple mediation effects

	Effect Size	SE	LLCI	ULCI
X1 (Social Support)				
Total	0.494	0.060	0.375	0.612
Direct	0.208	0.061	0.088	0.328
Total Indirect	0.286	0.043	0.207	0.377
Indirect effect (M1=Helplessness)	0.122	0.030	0.068	0.186
Indirect effect (M2= Internality)	0.163	0.032	0.108	0.230
(C1)	-0.04	0.043	-0.127	0.043
X2 (Health Literacy)				
Total	1.723	0.139	1.449	1.996
Direct	0.942	0.259	0.432	1.453
Total Indirect	0.780	0.217	0.373	1.218

17

Indirect effect (M1=Helplessness)	0.350	0.119	0.127	0.599
Indirect effect (M2= Internality)	0.430	0.143	0.161	0.720
(C1)	-0.080	0.148	-0.374	0.216

326 *Note: C1=Helplessness minus Internality.*

Comment 2. Correlation: What correlation did the authors use (Pearson or Spearman)? Have the authors checked the normality distribution before conducting the correlation analysis? If yes, please add.

Response 2: Thank you for your comments. All variables were normally distributed, so this study used Pearson for correlation analysis.

Change 2 (Page14 , line 281-282):

The main variables were normally distributed, so Pearson correlation analysis was performed to test the relationships among measured variables.

(Page12 , line 246-247)

If the data conformed to a normal distribution, the Pearson correlation coefficient was used to express the correlation between the two variables.

Comment 3. Missing data: How are missing data addressed? Did the authors use the complete case dataset? For example, as for the 24-item health literacy scale, if the respondent only answered 20 items, how to calculate the total score? This should be specified in the “Statistical analysis” section.

Response 3: Thank you for your comments. We checked the data integrity and processed missing values before analysis. Expectation-maximization or regression algorithms were used to produce estimates instead of missing values. Questionnaires with missing values exceeding 10% of the total items were considered invalid and given for exclusion.

Change 3 (Page 12, line 241-244):

Statistical analysis

We checked the data integrity and processed missing values before analysis. Expectation-maximization or regression algorithms were used to produce estimates instead of missing values. Questionnaires with missing values exceeding 10% of the total items were considered invalid and given for exclusion.

Comment 4. Re-structure the statistical methods: Some information on the multiple mediation analyses in the Results section should be moved upfront in the “Statistical analysis” section. Sth like “We used three-step regression models to examine Model 1 is to ...; Model 2 is to ...; Model 3 is to ...”

Response 4: Thank you for your comments. We have moved this section forward to the Statistical Analysis section.

Change 4 (Page 12-13, line 251-259):

This study used three regression models to examine the parallel mediators' role of helplessness and internality in the relationship between health literacy/social support and self-management, respectively. Model 1, with health literacy/social support as the independent variable and helplessness as the dependent variable. In model 2, health literacy/social support remained the independent variable, while internality was the dependent variable. Model 3 used health literacy/social support as the independent variable, helplessness, internality as mediating variables, and self-management as the dependent variable.

Some other comments are listed here:

- **Page 8 line 122-123:** "It has been suggested that learned helplessness is associated with high social support and health literacy" should be "It has been suggested that learned helplessness is associated with low social support and low health literacy."

Response: Thank you for pointing out our mistakes. We have changed the word "high" to "low".

Change (Page 7, line 130-131): It has been suggested that learned helplessness is associated with low social support and health literacy.

- **Page 11 Line 196-197:** "This instrument has been used in Chinese renal dialysis patients."

Reference is needed at the end.

Response: Thank you for your comment. We have added the citation.

Change (Page 10, line 214-215): This instrument has been used in Chinese renal dialysis patients²²

40.

[22] Song Y-y, Chen L, Wang W-x, et al. Social support, sense of coherence, and self-management among hemodialysis patients. *Western Journal of Nursing Research* 2022;44(4):367-74. doi: 10.1177/0193945921996648

[40]. Jiang H, Wang L, Zhang Q, et al. Family functioning, marital satisfaction and social support in hemodialysis patients and their spouses. *Stress Health* 2015;31(2):166-74. doi: 10.1002/smi.2541

• **Page 12 Line 218** : “In our previous study,” reference is needed here.

Response: Thank you for your comment. We have added the citation.

Change (Page 11-12, line 236-238): In our previous study, the scale showed a good internal consistency [(Cronbach’s α of 0.759), retest reliability (n=30, intragroup correlation=0.772), and split-half reliability [0.774]⁴⁴.

• **Table 1 on Page 14:** The ethnicity variable is missing here. The authors highlight “ethnicity” is an important variable in the Limitation section. Why? Does this mean only “Han” ethnicity was recruited? Is there any study that found ethnicity will influence the findings of relationship between health literacy/social support, helplessness and internality, and self-management behaviours?

Response: Thank you for your comments. There are 56 ethnicities in China, and given the cost of research and time, this study has only been conducted in one southern city in China, which is one of the limitations of this study. In addition, the existing studies did not verify the moderating effect of ethnicity variables on the mechanism. In the future, we hope to further explore this mechanism by including more HD patients other than Han ethnicity through a multicenter collaboration.

Change: N/A.

• **Table 1 on Page 14:** The education level does not include “high school”? Maybe change the “Middle school” to “Secondary school”?

Response: Thank you for your comment. We did mean to include the middle and high school. Therefore, based on your suggestion, we have changed the word "middle school" to "secondary school"

Table 1 Descriptive statistics of socialdemographic characters (n=239)

Variable	n	%
Gender		
Male	143	59.8
Female	96	40.2
Age (years)		
18-44	46	19.2
45-59	100	41.8
≥60	93	38.9
Education level		
Elementary school	26	10.9
Secondary school	126	52.7
University and above	87	36.4
Employed status		
On duty	10	4.2
Part-time job	37	15.5
Off duty	192	80.3
Protopathy		
Hypertension	52	21.8

Change:

• **Page 18 Line 294-316:** The first paragraph in the Discussion section needs to be revised: It should be the key summary of findings, corresponding to the aforementioned three hypotheses. Also, there is lack of discussion on the hypothesis 1 in the Discussion section. Currently, the authors focuses on more discussions on hypothesis 2 and hypothesis 3.

Response: Thank you for your advice. We have rewritten the first paragraph of the Discussion section.

Change (Page 18-20, line 328-380):

DISCUSSION

Based on the COM-B model, this study jointly explored potential factors influencing self-management in MHD patients at three levels: opportunity, competence, and motivation, and elucidated the pathways among the factors by constructing a theoretical model. Our findings confirm the COM-B model's applicability in health-related behavioral research among MHD patients. First, health literacy/social support, as important competence and opportunity factor for MHD patients, was positively and negatively associated with internality and helplessness, respectively, and positively associated with self-management. Second, learned helplessness, an important motivational factor influencing individual behavior, and its two major systems, internality and helplessness, were positively and negatively associated with self-management in MHD patients, respectively. Furthermore, our findings suggest that health literacy/social support influences self-management through internality and helplessness as mediating factors.

Correlation between health literacy/social support and learned helplessness

This study first verified the correlation between health literacy and the component of learned helplessness (internality/helplessness). According to the knowledge-attitude-practice theory⁴⁶, patients with some knowledge of kidney disease and dialysis rated their self-behavior better and had stronger beliefs about their illness. They could make rational judgments and perceptions when encountering negative disease-related events, accompanied by a lower sense of helplessness, which was verified in this study. Conversely, patients with higher internality are more likely to acquire disease-related knowledge through various means, which is vital for patients' self-management. Those patients who were chronically helpless lacked the motivation to consistently learn self-care knowledge and skills. This point echoes the findings of a previous qualitative study of ours⁴⁷. Therefore, successful renal disease education should provide knowledge and increase motivation. Healthcare professionals should also emphasize patient internality and empowerment to help patients maintain a positive mindset and long-term intrinsic motivation to face their disease.

Next, this study verified the close relationship between internality/helplessness and social support. According to Shahriari⁴⁸, social support as an important external resource can lead to significant psychological changes in patients. This study reflects two critical variables of internality and helplessness. On the one hand, a lack of social support can induce helplessness in patients, mainly depression, social avoidance, and abandonment. On the other hand, adequate social support can help

patients expand access to social resources, increase internality in coping with illness, and reduce helplessness, which is similar to the findings of Wang⁴⁹ and Wu⁵⁰. It is worth noting that Jansen⁵¹ pointed out that excessive support given to MHD patients may affect their internality. In this regard, assisting patients to build an appropriate social support environment is essential to enhance their internality and temporarily escape from helplessness.

• **Page 19 Line 309-313:** The definitions of helplessness and internality (“Learned helplessness was divided into ... their life.”) should be moved upfront in the Introduction “Learned helplessness” section.

Response: Thank you for your comments. We have migrated this section to the introduction on learned helplessness in the **Background section**.

Change (Page 6-7, line 126-129):

Learned helplessness

.....Learned helplessness was divided into two core components. Helplessness is the individual's perception that their behavior cannot impact the outcome²⁹. Internality refers to the individual's perception that they believe that they are the most dominant force dominating their life.

• This study treated internality as one important mediating variable based on the learned helplessness theory, which is similar to self-efficacy (the author also mentioned a bit in the introduction) and locus of control. It would be good to add some discussion around their differences and roles in self-management. The authors already cited some good references (ref 6 “self-efficacy”; ref 16 “self-efficacy”; ref 51 “locus of control”; ref 53 “locus of control”)

Response: Thank you for your comments. Internality is derived from Rotter's Locus of control theory, which means that internality is a type of Locus of control. Therefore, we have only discussed the differences between self-efficacy and internality for self-management.

Change (Page 18-19, line 351-341):

It is worth noting that internality, the core system of LH, has similarities with self-efficacy in that both can directly or indirectly influence self-management. However, the two are presented in different theoretical contexts, determining the intervention's core. Self-efficacy, derived from Bandura's social learning theory⁴⁵, refers to an individual's subjective evaluation of his or her abilities, driven by personal expectations and goals. In this context, the core of self-management interventions is to help patients set reasonable goals and gradually develop their self-confidence. Internality originates from Rotter's Locus of control theory³¹, which refers to a stable way of attributing the outcome of events internally after individuals have learned through experience. The core of its health behavior intervention is to guide patients to rational attribution, reduce helplessness and enhance motivation.

Reviewer: 1

Competing interests of Reviewer: I declare no competing interests.

Reviewer: 2

Competing interests of Reviewer: No competing interests.

Reviewer: 3

Competing interests of Reviewer: None

VERSION 2 – REVIEW

REVIEWER	Bayuo, Jonathan Hong Kong Polytechnic University, School of Nursing
REVIEW RETURNED	01-Apr-2023
GENERAL COMMENTS	Many thanks to the authors for thoughtfully addressing all the comments raised.

VERSION 2 – AUTHOR RESPONSE